# Predicting C/N$_0$ as a Key Parameter for Network RTK Integrity Prediction in Urban Environments

**Ali Karimidoona** *  **and Steffen Schön**

Institut für Erdmessung, Leibniz Universität Hannover, Schneiderberg 50, 30167 Hannover, Germany; schoen@ife.uni-hannover.de

* Correspondence: karimidoona@ife.uni-hannover.de; Tel.: +49-511-762-3452

**Abstract:** Autonomuous transportation systems require navigation performance with a high level of integrity. As Global Navigation Satellite System (GNSS) real-time kinematic (RTK) solutions are needed to ensure lane level accuracy of the whole system, these solutions should be trustworthy, which is often not the case in urban environments. Thus, the prediction of integrity for specific routes or trajectories is of interest. The carrier-to-noise density ratio (C/N$_0$) reported by the GNSS receiver offers important insights into the signal quality, the carrier phase availability and subsequently the RTK solution integrity. The ultimate goal of this research is to investigate the predictability of the GNSS signal strength. Using a ray-tracing algorithm together with known satellite positions and 3D building models, not only the satellite visibility but also the GNSS signal propagation conditions at waypoints along an intended route are computed. Including antenna gain, free-space propagation as well as reflection and diffraction at surfaces and vegetation, the predicted C/N$_0$ is compared to that recorded by an Septentrio Altus receiver during an experiment in an urban environment in Hannover. Although the actual gain pattern of the receiving antenna was unknown, good agreements were found with small offsets between measured and predicted C/N$_0$.

**Keywords:** GNSS; network RTK; urban navigation; integrity monitoring

## 1. Introduction

The growing number of applications of autonomous vehicles and the fact that driving autonomous vehicles in any level of automation is a safety critical application necessitate very precise and accurate determination of the positions of such vehicles, with a high level of integrity. The levels of autonomy range from zero level (no automation) to level five (fully automated without human interaction or even attention) [1]. Finding the precise position of the vehicle (i.e., vehicle localization) is a fundamental part of the next steps of autonomous vehicle driving which further includes perception, prediction, planning and control [2].

The only sensor that provides an absolute position of objects is the Global Navigation Satellite System (GNSS) [3,4]. The signals transmitted from these satellites in L-band are strongly attenuated due to the long distance (around 20,000 km) of the satellite from the Earth's surface [5]. In addition, atmospheric refraction, satellite and receiver clock errors and orbit errors impact the observation error budget, to just name some of the issues [6].

The error contributions which are common or very similar for two receivers at different locations or between two satellites observed by one receiver can be eliminated or at least largely reduced by differencing, e.g., double differencing (DD) [7]. Examples of such errors are delays by ionospheric and tropospheric refraction, satellite orbit errors, and satellite and receiver clock errors. Real-time kinematic (RTK) positioning solutions rely on very precise carrier phase observations (mm noise level) which are two orders of magnitude (i.e., factor 100) more precise than pseudorange observations. In Network RTK (NRTK), a network of several reference stations is producing and distributing corrections for the

distance-dependent error sources [8]. NRTK solutions deliver cm level accurate position results in optimum situations. If the number of observations is not high enough or if their quality is not good enough, the cm precision is not guaranteed and the solution degrades to the dm or even m level. The availability of sufficient dual-frequency phase observations with high $C/N_0$ is a key factor for high-precision RTK positioning. However, location-specific errors remain, which are not common between receivers nor between satellites, namely multipath or non-line-of-sight (NLOS) signal reception [9]. These kinds of errors are more severe in urban environments where GNSS signals are interacting with different obstacles such as buildings, bridges, trees, utility poles, etc. [10,11].

For mitigating NLOS and multipath effects, different approaches have been used, namely antenna and receiver design, signal processing techniques, navigation-based techniques, postprocessing and 3DMA (three-dimensional-mapping-aided) techniques [12], which include ranging based and shadow matching, i.e., comparing the $C/N_0$ to nominal values to determine blockages. Here, our focus is on the 3DMA approach. The work conducted at University College London (UCL) in collaboration with Spirent company [13,14] shows the satellite signal can be received rather than LOS, in diffracted, reflected and multipath situations. "SPRING", a simulating software developed at a French space agency (CNES) in collaboration with Thales Services company, is a toolbox which considers the reflection and diffraction of the signal, and [15] shows the NLOS effect on positioning error. The physical and statistical properties of the multipath are considered in the "SCHUN" simulator developed at French national aerospace research laboratory (ONERA) [16]. Researchers at Gustave Eiffel University performs ray-tracing algorithms to forecast the satellite visibility [17] and to correct the NLOS observations for better positioning results [18–20]. 3DMA in the work of IPN Laboratory of Hong Kong Polytechnic University [21] is used to weigh the NLOS observations rather than exclude them. Other works from this group consider 3DMA GNSS positioning in urban environments [22–24]. Commercial product "Sim3D" by Spirent [25] simulates the $C/N_0$ considering multipath and NLOS situations by incorporating different 3D objects as desired by the user, e.g., building, tree, car, etc. The 3DMA technique has been used in the "PosNav" research group of the Institut für Erdmessung at Leibniz University Hannover to investigate the NLOS and multipath effect in urban environments [11,26–28]. Current work basically takes advantage of [26], in which the multipath effects are modelled specifically for the carrier phase by developing compact expressions.

In order to guarantee a navigation service for a specific trajectory with predefined quality (e.g., accuracy, integrity), performance prediction can be incorporated into the path planning step. The overall work aims to predict integrity, namely position error and protection level, in NRTK positioning useful for autonomous driving in urban environments [29,30]. For this purpose, it is necessary to first identify the driving parameters for integrity of the RTK positioning and predict their evolution for different trajectories, environments and satellite sky distributions.

By predicting the signal availability and $C/N_0$, we can put one step further towards predicting integrity. Prediction of satellite visibility in urban environments has become possible due to 3D building models of the cities. For cities located in mountainous areas, the terrain model should also be considered. This technique is known as 3D-mapping-aided (3DMA) GNSS [12]. For predicting the $C/N_0$, in addition to signal visibility, the status of visibility should also be known. The visibility status can be line-of-sight (LOS), non-line-of-sight (NLOS), multipath or blocked [31]. The NLOS situation itself can be divided into reflection or diffraction. The signal propagation in space and attenuation interacting with the environment for any visibility status until being received by the receiving antenna should be considered (Section 4).

The remainder of the paper is structured as follows. In Section 2, the real-world kinematic experiment is explained, which has been conducted to gather real observations from geodetic-grade GNSS receivers. In Section 3, we see that the solution quality can deviate from the ideal RTK solution underlining the importance of $C/N_0$. In Section 4, the

signal strength prediction is investigated. In addition to a ray tracing algorithm, information is given regarding signal propagation in space and also interaction of the signal with the environment to calculate the corresponding fading of the signal in each visibility status. In Section 5, results from $C/N_0$ prediction are shown and discussed. Conclusions are given in the last section.

## 2. Experiment

A kinematic experiment was conducted on 29 July 2021, in which four different RTK receivers, namely Leica GS18 T, Trimble R12i, Septentrio Altus NR3 and an integrated antenna with a u-blox ZED-F9P receiver, were used (Figure 1b). In addition to the RTK receivers, one navigation-grade IMU (iMAR iNAT-RQT-4003) (Figure 1c) was installed inside the vehicle to compute a precise reference trajectory [32]. GNSS RTK receivers have internal GSM modems and are capable of connecting to World Wide Web to receive the differential corrections from a reference station or a network of reference stations (N-RTK) in an Ntrip format [33]. In this experiment, the Network RTK correction stream is acquired by the VRS mount-point of SAPOS® service delivering GPS and GLONASS corrections in the time of experiment in Hannover, supported by the Lower Saxony State office for Geoinformation and surveying (LGLN) [34]. When referring to real data in the remainder of this paper, we are using here exemplarily the real data from the Septentrio Altus NR3 (Figure 1, the receiver on the far right in Figure 1b). An eight-shaped trajectory (Figure 1a) was driven for twelve rounds divided into two sections of six rounds separated by two hours in order to cover various satellite sky distributions. Each round of the eight-shaped trajectory was almost 1 km long and took approximately 5 min to drive. The GNSS receivers log data in 10 Hz sample rate and the rate for IMU is 400 Hz.

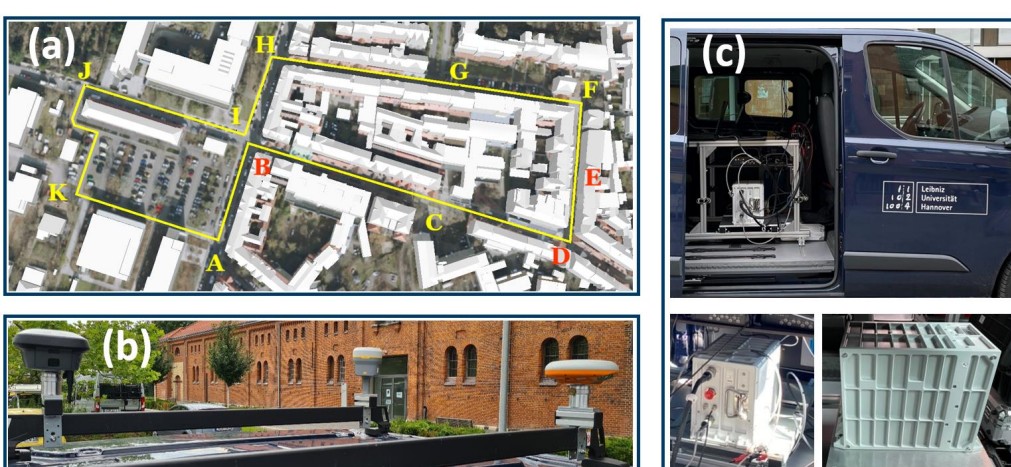

**Figure 1.** Overview of the kinematic experiment. (**a**) Eight-shaped trajectory driven in an urban environment of Hannover near Leibniz University campus; A to K are the way-points which show the direction of the drive; this trajectory includes parts with quite open sky situations in the parking lot (J to A), and some parts with very difficult sky visibility (D to F). Different colors of way-points are merely chosen for better visibility of the image. (**b**) Experimental set-up with RTK receivers mounted on top of the van. (**c**) Installation of the navigation-grade IMU (iMAR iNAT-RQT-4003) inside the van.

## 3. Analysis of the Positioning Solution and Observation Data Quality

The availability of carrier phase signals (in two frequencies) and correction streams as well as the signal quality determine the solution type, namely fixed, float, code and navigated solutions, which are introduced in this section.

### 3.1. Solution Type

For the *fixed solution* case, the carrier phase ambiguity is solved into an integer value which is the nature of this parameter [35]. This is not always possible due to the bad observation quality or geometric weakness to decorrelate the ambiguities completely. In such cases, the ambiguities are estimated as floating point numbers. This solution is called *float solution*. If there are too few carrier phase observations available to solve the ambiguities or the quality of the carrier phase observations is extremely low, the positioning is a *code solution* based on pseudorange observations only. In this case, the pseudorange corrections are used to improve the positioning solution. The code solution is also known as DGNSS or DGPS [36]. Furthermore, if the quality of the pseudoranges is not good enough to use the pseudorange corrections, the filter calculates a standalone solution, i.e., using only the pseudorange observations without any corrections. In this case, the positioning results from a *navigated solution*.

In Figure 2, the horizontal components of the position solutions from the Altus receiver in the fifth round of our experiment are exemplarily depicted. Solution types are color-coded regarding the legend. The solutions named float, code and navigated are depicted by circles whose center is the position solution and the radii reflects the 2D coordinate quality (2DCQ) of the respective epoch. A circle with a radius of 3 m is shown in the legend for float, code and navigated solutions, which indicates the scale of the circles in the figure. The minimum and maximum 2DCQ of the fixed solutions are 0.03 m and 0.1 m, respectively. The arrows indicate the direction of the vehicle driving, and way-points A to K help to reference the geometric situation, cf. also Figure 1a.

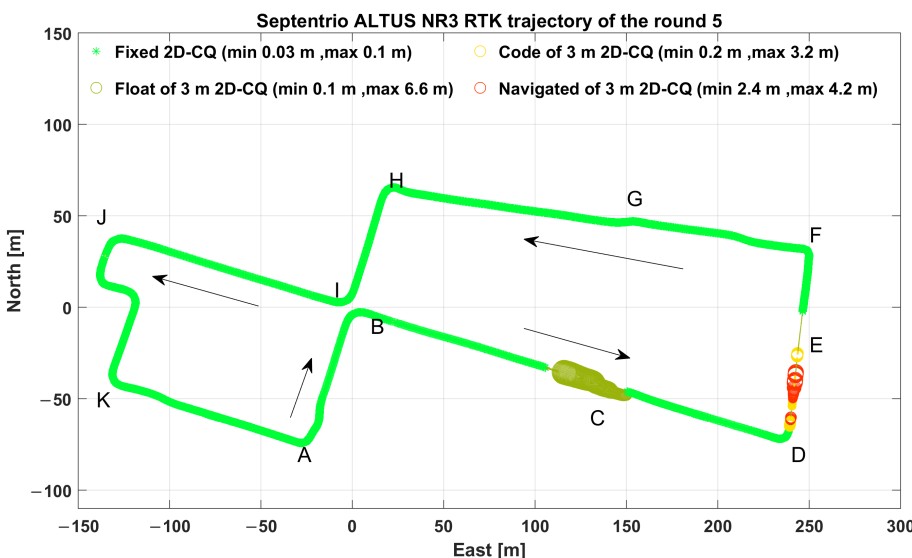

**Figure 2.** Estimated trajectory from the Altus receiver in the fifth round of the experiment. Different colors representing the four solution types. Fixed solutions are depicted in light green stars. The other solutions are depicted as circles which are centered at the horizontal solution and their radii show the 2D coordinate quality. In the legend, the circles have a radius of 3 m as a scale for the circles in the figure. The arrows indicate the direction of the driving. The way-points A to K are as defined in Figure 1a.

In Figure 2, the segment between way-points D and F is the most challenging part of the trajectory, which is a narrow north–south-oriented street with mostly four-story buildings on both sides and also at both ends (Figure 1a). The fixed solution is not maintained in this part anymore. The cellular modem status recorded by the receiver (Septentrio Altus NR3) shows that the connection has not been interrupted and that RTK corrections were continuously received. Therefore, referring to the navigated solution is not related to the availability of the corrections, but rather the quality of the signals received.

### 3.2. Number of Available Dual-Frequency Signals

One key factor to enable a fixed solution is the strength of the employed mathematical model, which depends on the receiver-satellite geometry or dilution of precision (DOP, ADOP), the number of observed signals, the observation period, the observation precision and the dynamics of the object [37].

In addition, the availability and continuity of dual-frequency carrier phase observations play an important role, here investigated for GPS (GL1C/GL2L or GL1C/GL2W) and GLONASS (RL1C/RL2C) observed by the Altus receiver.

Figure 3 shows the number of dual frequencies for GPS, GLONASS and the sum for every epoch during the fifth round. The color codes indicate the solution status as shown in Figure 2. The way-points indicate the position of the receiver, cf. Figure 2. Different saturations from light to dark specify GLONASS dual-frequency availability, GPS dual-frequency availability and the total number, respectively, as shown in the legend. It can be seen that the least total number (five) occurs before way-point E, where the solution status is navigated. The numbers of the dual-frequency phase observations are clearly in lower cases in float, code and navigated solutions compared to those in fixed ones. Part D to F of the trajectory is a difficult situation for satellite visibility (cf. Figure 1a).

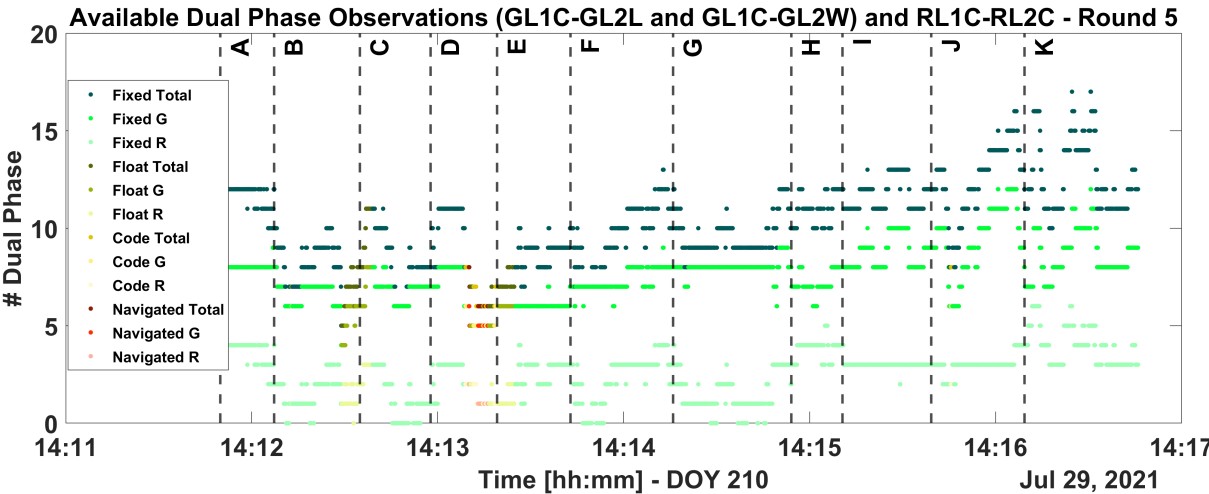

**Figure 3.** Number of dual-frequency phase observations for GPS (G) plus GLONASS (R) color-coded by solution status. The way-points A to K are as defined in Figure 1a.

Figure 4 shows the histogram of the dual-frequency phase availability for the four solution statuses in all twelve rounds. For the status fixed solution (Figure 4a), there are at least four dual-frequency phase observations available, with two for GPS and two for GLONASS. But this case is really rare, with only two instances (two epochs). In sixteen cases, five dual-frequency phase observations (three G + two R) have occurred. The replication for six (three G + three R) increases to nearly 200 occurrences. The probability to have a fixed solution is the greatest for six dual-frequency GPS observations. There are only four instances for float solutions with two G and two R dual-frequency observations (Figure 4b). Mostly, float solution are reported with four G and three R dual-frequency observations. For the solution-type code, we can see that there are solutions even with two dual-frequency observations. The most probable case is four G and two R dual-frequency observations (Figure 4c). For a higher number of dual-frequency observations, it is less probable that the solution go to navigated, but still with seven (five G + two R) available dual-frequency phase observations there are navigated solutions (Figure 4d). It is more probable for navigated solutions to have only one G and one R dual-frequency observations.

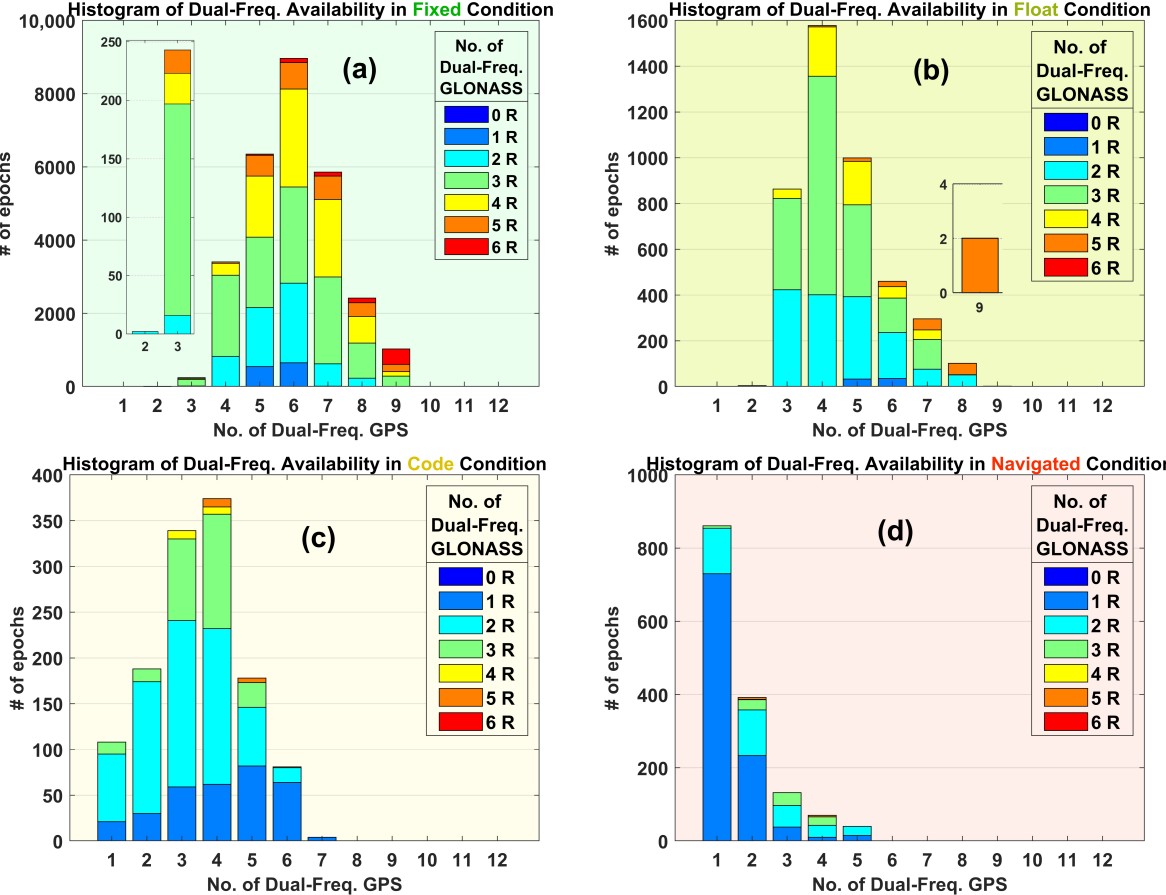

**Figure 4.** Histogram of available dual-frequency (L1 and L2) phase GPS (G) and GLONASS (R) signals for four types of position solutions: (**a**) fixed, (**b**) float, (**c**) code and (**d**) navigated in all twelve rounds.

### 3.3. Signal Strength

Signal strength in the context of GNSS is usually expressed as carrier-to-noise-density ratio, $C/N_0$. This parameter is one of the observations reported by the receiver. It is a key parameter for evaluating the quality of the signal and thence affecting the positioning. Figure 5 shows the GPS and GLONASS $C/N_0$ values of Round 5, where the color bar at the bottom shows the solution statuses as introduced earlier in this section (cf. Figure 2). Before way-point C, as can be seen from the color bar, the solution proceeds to float type. At this moment, we can see a sudden drop in very high values that results in changing the solution type. Additionally, after way-point D, the solution priceeds intermittently to code, float and navigated types. It can be seen that at this time, there is a sudden drop in the signal strength for some of the satellites. The minimum $C/N_0$ observation (GS1C) is 17.4 dB-Hz. Looking at the carrier phase observations, we can detect a threshold of 22 dB-Hz for tracking GL1C.

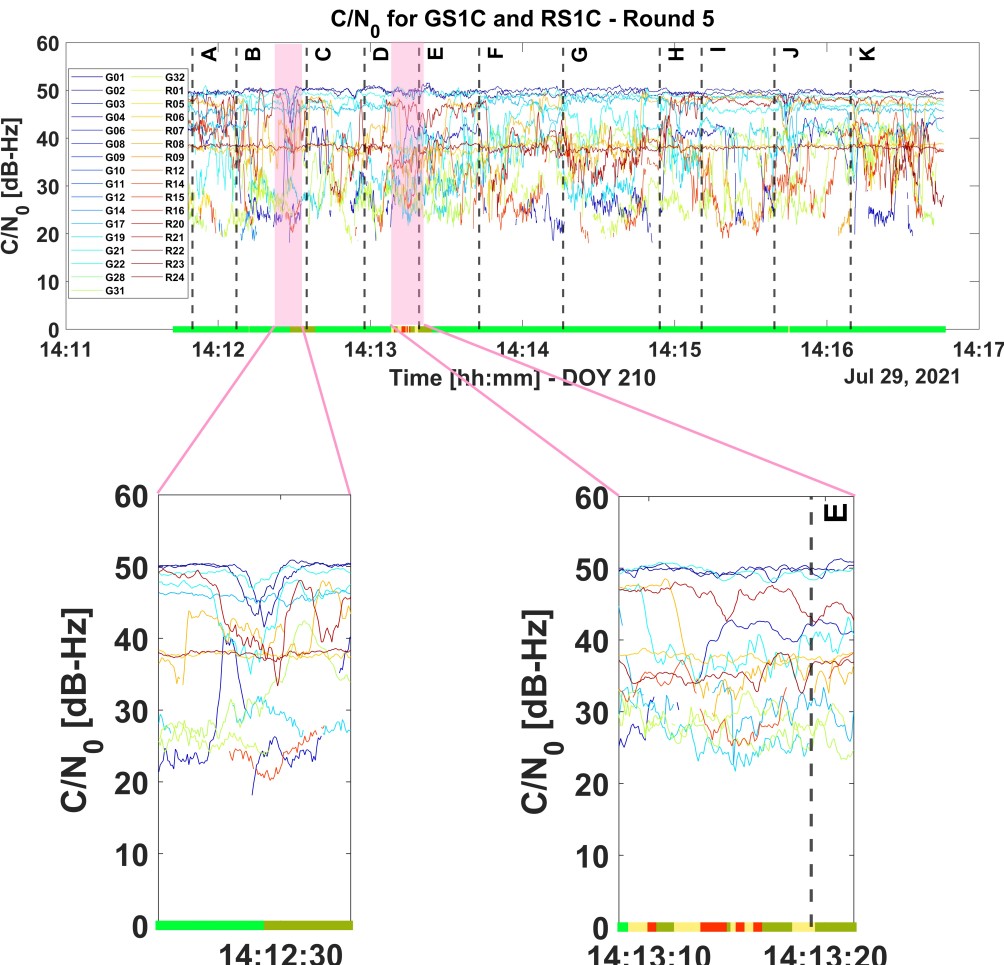

**Figure 5.** Signal strength variation for GPS and GLONASS L1 signals shown for Round 5. The color code in the bottom of the figure indicate the solution type: fixed (green), float (olive), code (yellow) and navigated (red), cf. Figure 2. The way-points A to K are as defined in Figure 1a.

## 4. Predicting Signal Strength

In the previous section, we saw that at epochs where the positioning solution is not fixed and becomes float, code or navigated, there is a degradation in the $C/N_0$ of some satellites, hence this can suggest a correlation between $C/N_0$ and the quality of the positioning. Looking at $C/N_0$ helps us to better detect the challenging points and can function as an indicator for our prediction.

In the following subsections, we see that, first, we need to predict the satellite visibility, and after that evaluate the link budget based on the status of the signal.

### 4.1. Classification of the Signal Reception Conditions by Ray-Tracing

A signal (ray) may fall into one out of four categories as shown in Figure 6 [27]. Line-of-sight (LOS) is the case where the satellite is only directly visible from the antenna position. Non-line-of-sight (NLOS) reception happens if the signal impinges the antenna in a way other than direct path, which can be either a reflection or diffraction. In this study, by NLOS we mean specular reflection only. Diffraction events are mentioned separately. If the signal comes to the antenna both along the direct path and also from a reflection, then it is called a multipath. In a blocked situation, the signal is halted from reaching the antenna by any means, usually a large block of building or mountain. It should be mentioned that here, only single reflections are considered, so the case of blocked signals may also contain situations of multiple reflections. Diffraction can occur in all the four cases (LOS, NLOS, MP and blocked).

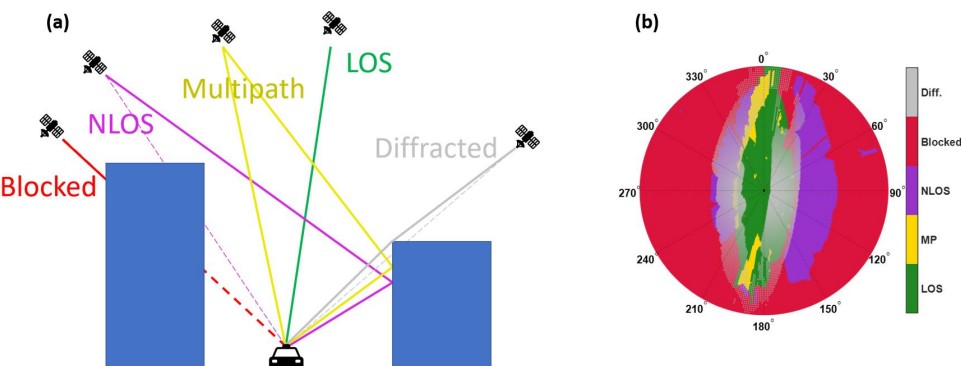

**Figure 6.** Classification of GNSS signal reception conditions. (**a**) Satellite signals may experience one of the four situations: LOS, NLOS, multipath or blocked. Diffraction can occur in addition to any of the aforementioned cases. (**b**) Exemplary skyplot with corresponding visibility status for every azimuth and elevation. This skyplot belongs to a point in the middle of the north–south oriented street near way-point E (cf. Figure 1a).

This signal reception classification has become possible due to 3D city models. The precise geometric information which is provided by the coordinates of the building models is divided into several polygons. The standard CityGML (Geography Markup Language) Level of detail 2 (LoD2) which is used here includes the details of the ridge or dormer rooftops of the buildings which are typical in German cities. In this study, the 3D building model Level of detail 2 (LoD2) provided by the city Hannover is used. The 3D city model is divided into 1 km by 1 km, gml files. Based on the coordinates of the experiment, a large enough area is selected to include all the buildings that might be influential in the ray-tracing. A nearby skyscraper which is not included in the selected area is additionally considered because although it is quite far, it has direct influence on the signals. Coordinate transformations are performed as indicated in [38].

Satellite coordinates and their temporal evolutions are known for every constellation by almanacs in advance, while more precise information is broadcast in real time from the satellites through navigation messages.

The last type of information needed for the classification is the position of the moving receiver. For prediction purposes, the user position can be obtained from a path planning, assuming that a specific trajectory should be driven a given moment in time. Results of [28,39] show that the ray-tracing classification is preserved for a road section of few meters. In this study, in order to ease the direct comparison with the measurements, the coordinate time series of the reference trajectory are used that were obtained by tightly coupling the GNSS and IMU data.

Knowing the coordinates of the satellite and the receiver antenna, the ray tracing is performed [40]. For diffraction, the third Fresnel ellipsoid is considered as the obstruction criteria, cf. [41].

*4.2. Predicting Satellite Visibility*

In Figure 7a, the signal visibility for GPS is shown as predicted by the ray-tracing algorithm. The color codes indicate the signal reception classes, cf. Figure 6. Green is LOS, purple is NLOS, yellow shows multipath and gray indicates diffraction. Here, the gray color implies that the satellite is in a blocked situation but the signal arrives from diffraction. The small red vertical bars indicate the start and end epochs of signal interruptions.

The continuity of the real code and phase signals for GPS constellation are shown in Figure 7b,c respectively. While PRN 14 has been predicted mostly in a diffracted mode, it has not been observed in real data, neither in code nor carrier phase. Maybe this is due to the fact that for the diffraction calculation, the third Fresnel ellipsoid has been considered. This can imply that probably lower-order Fresnel ellipsoids are more realistic to be used. There are many interruptions in the code signal in PRNs 31 and 32; but the carrier phase

signal has been observed less for these two satellites. This can imply the difficulty of maintaining the carrier phase in more interrupted signals rather than code.

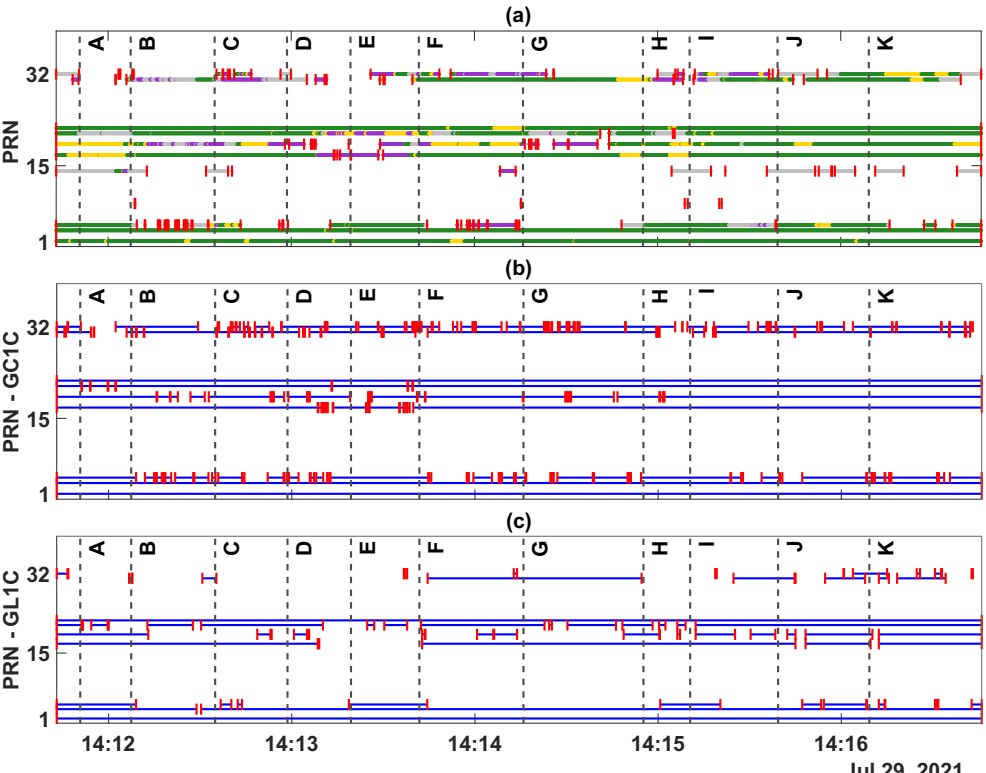

**Figure 7.** Predicted and observed GPS satellite visibility exemplarily shown for the fifth round of the experiment: (**a**) The predicted GPS visibility, the LOS, NLOS, MP and diffracted signals are depicted in green, purple, yellow and gray color, respectively. (**b**) The real visibility of code GC1C signal. (**c**) The real visibility of the GL1C signal. The real data are exemplary from the Altus receiver. Short vertical red lines indicate the signal start or interruptions. The way-points A to K are as defined in Figure 1a.

Figure 8a–c, depict the predicted, real code and real phase visibility, respectively, for GLONASS system. As it can be seen in the prediction part, PRNs 5, 7, 8, 14, 15, 21 and 22 are predicted to be visible, while in reality, PRNs 5, 7, 14, 21 and 22 for code and 7, 14, 21 and 22 for phase are observed.

The confusion matrix between predicted satellite visibility and real code and phase observations for GPS and GLONASS for the fifth round are reported in Table 1. This table shows the extent to which the predicted visibility complies with real data. There are two values in each cell. The first one without parentheses is the percentage of the total number of satellites meaning 32 for GPS and 24 for GLONASS, while the value in parentheses indicates the percentage referring only to the predicted satellites meaning PRNs 1, 3, 4, 14, 17, 19, 21, 22, 31, and 32 for GPS and PRNs 5, 7, 8, 14, 15, 21, and 22 for GLONASS. As an example, in 18.69% of the epochs of the fifth round (3031 epochs), the predicted visibility is the same as the real phase visibility for the GPS constellation (all 32 satellites). This value for only the predicted visible satellites is 59.81%. These values can be visible in the upper right panel of Table 1. In this panel, considering all satellites, 90.56% (18.69 + 71.87) of the epochs are correctly predicted (true positive and true negative); while considering the predicted satellites, a total of 77.61% (59.81 + 17.80) of the epochs are correctly predicted. This means that the total visibility status of the satellites are well predicted, but in the details of the visibility for predicted PRNs, the situation is more difficult.

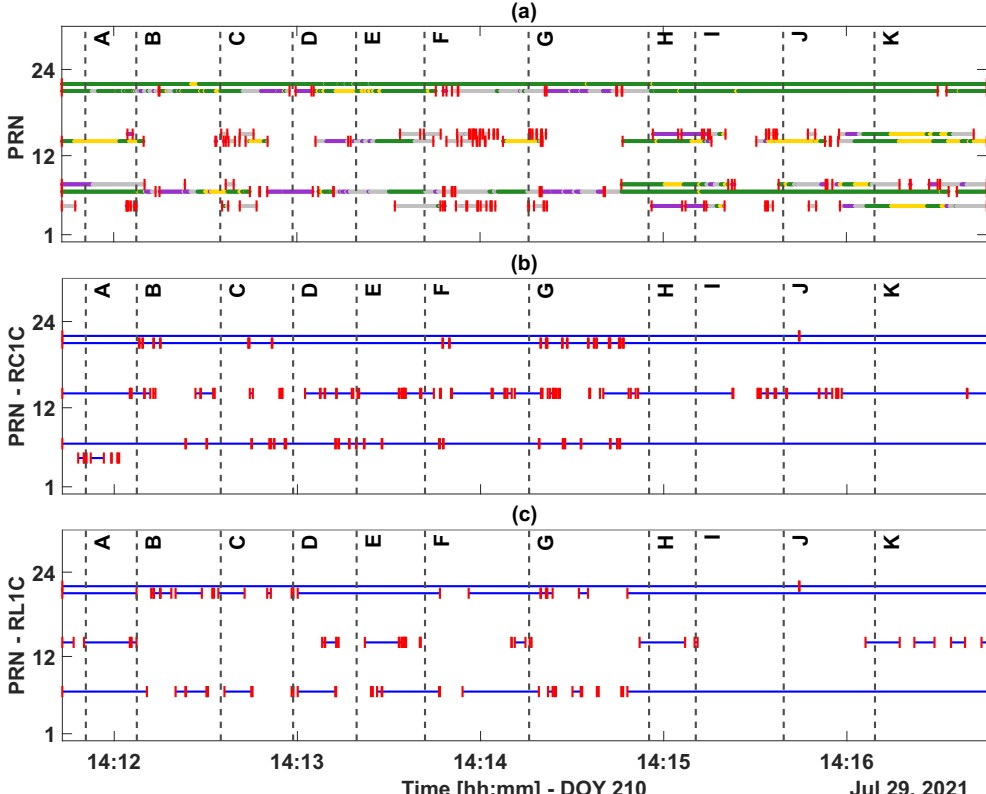

**Figure 8.** Predicted and observed GLONASS satellite visibility exemplarily shown for the fifth round: (**a**) The predicted GLONASS visibility, the LOS, NLOS, MP and diffracted signals are depicted in green, purple, yellow and gray color, respectively. (**b**) The real visibility of code RC1C signal. (**c**) The real visibility of the RL1C signal. The real data are exemplary from the Altus receiver. Short vertical red lines indicate the signal start or interruption. The way-points A to K are as defined in Figure 1a.

**Table 1.** Confusion matrix of the predicted and real visibility status. The green color indicates true positive and true negative, the red color indicates false positive and false negative. Numerical values are in percent for all 32 GPS and 24 GLONASS satellites in all epochs of the fifth round. The values in parentheses show the percentages only for predicted satellites.

|  |  | Code | | |  | Phase | | |
|---|---|---|---|---|---|---|---|---|
| **GPS** | **%** | Real Visible | Real Not Visible |  | **%** | Real Visible | Real Not Visible |  |
|  | Predicted Visible | 23.74 (75.96) | 4.33 (6.05) |  | Predicted Visible | 18.69 (59.81) | 9.38 (22.20) |  |
|  | Predicted Not Visible | 2.45 (7.83) | 69.48 (10.16) |  | Predicted Not Visible | 0.06 (0.19) | 71.87 (17.80) |  |
| **GLONASS** | **%** | Real Visible | Real Not Visible |  | **%** | Real Visible | Real Not Visible |  |
|  | Predicted Visible | 14.57 (49.96) | 5.10 (17.48) |  | Predicted Visible | 11.75 (40.29) | 7.92 (27.1) |  |
|  | Predicted Not Visible | 1.00 (3.42) | 79.34 (29.15) |  | Predicted Not Visible | 0.15 (0.51) | 80.18 (32.05) |  |

### 4.3. Signal Power in Line-Of-Sight Conditions

In a LOS situation, the signal attenuation obeys the Friis transmission equation [42],

$$P_r/P_t = \mathrm{A}_t A_r/d^2\lambda^2 = G_t G_r \left(\frac{\lambda}{4\pi R_{txrx}}\right)^2, \tag{1}$$

where $P_t$ is the power fed into the transmitting antenna which is 27 watt (or 14.3 dB W) which is mentioned in the GPS specifications, while in the real-world, GPS space vehicles transmit up to 4 dB more power [43]. The power transmitted by the satellites varies by different space vehicles and decreases with age, which can be, to some extent, compensated by examining the current maximum signal strength of each satellite [44]. $P_r$ is the available power at the output terminals of the receiving antenna. $G_t$ and $G_r$ are the gains of the transmitting and receiving antennas, respectively. The wavelength of the signal is denoted by $\lambda$ and $R_{txrx}$ is the distance between satellite and receiver antennas. The antennas should have the same polarization and the signal travels in unobstructed free line of sight between the antennas.

The term $A_{los} = (\lambda/4\pi R_{txrx})^2$ is called free space loss. In Equation (1), the atmospheric loss and other possible losses, e.g., cable loss, are ignored. Atmospheric loss is in the order of 0.035 dB for zenith angle and increasing with an order of ten times for low elevation satellites [45].

The GPS signals are right-hand circularly polarized [46]. Circular polarization can be interpreted as a combination of horizontal and vertical linear polarizations. Different approaches exist to show the polarization of the signal, among them the *Jones* vector is an easy-to-implement representation [47]. The Jones vector for the receiving antenna is defined as

$$\mathbf{e}_{rx} = \frac{1}{\sqrt{2}}\begin{bmatrix} G^{rx}_{rhcp_{AoA}} + G^{rx}_{lhcp_{AoA}} \\ -jG^{rx}_{rhcp_{AoA}} - G^{rx}_{lhcp_{AoA}} \end{bmatrix}, \tag{2}$$

where rhcp is the right-hand circular polarization and lhcp is the left-hand circular polarization for the angle of arrival (AoA), respectively. Analogously, the Jones vector for the transmitting antenna for the angle of departure (AoD) can be defined. Therefore, the signal received in an LOS situation is

$$S_{LOS} = P_t \mathbf{e}^H_{rx}\mathbf{e}_{tx} A_{los}e^{-j\alpha}, \tag{3}$$

and thus

$$P_{LOS} = |S_{LOS}|. \tag{4}$$

### 4.4. Signal Power in Non-Line-Of-Sight and Multipath

When a signal strikes a surface, the reflection can be specular or diffuse. In the diffuse mode, which occurs at rougher surfaces, the signal is scattered into many rays with different angles. In this case, the power of the coming signal is divided into many parts, and it is quite improbable that a signal can reach the receiving antenna. In the specular reflection case, which happens at smoother surfaces, the signal is reflected as it happens in a mirror. The power of the signal is partly absorbed by the reflecting surface. If this kind of reflected signal arrives at the receiving antenna in addition to the LOS, a multipath situation is occurring and the reflected signal is the multipath component (MPC). If this reflection is the only signal arriving at the receiver without any direct signal, it is called NLOS.

$$S_{MPC} = P_t \mathbf{e}^H_{rx}\mathbf{H}\mathbf{e}_{tx} A_{MPC}e^{-j\alpha}, P_{MPC} = |S_{MPC}|, \tag{5}$$

where $\mathbf{e}$ is the Jones vector (Equation (2)) with superscript H indicating the Hermitian conjugate. Matrix $\mathbf{H}$ (Equation (6)) consists of Fresnel reflection coefficients $\Gamma_H$ for horizon-

tal and $\Gamma_V$ for vertical polarization components (Equations (7) and (8)) to account for the specular reflection process, and the two rotation matrices to align the polarization ellipses of the two antennas [48]. $A_{MPC}$ is the free space loss for the multipath reflected component. The loss due to a quite large extra path delay (EPD) of 450 m of the reflected signal is 0.999975 and is, in practice, negligible, so it can be considered equal to $A_{LOS}$ [49].

$$\mathbf{H} = \begin{bmatrix} cos(\psi_{rx}) & sin(\psi_{rx}) \\ -sin(\psi_{rx}) & cos(\psi_{rx}) \end{bmatrix} \begin{bmatrix} \Gamma_H & 0 \\ 0 & \Gamma_V \end{bmatrix} \begin{bmatrix} cos(\psi_{tx}) & sin(\psi_{tx}) \\ -sin(\psi_{tx}) & cos(\psi_{tx}) \end{bmatrix}, \tag{6}$$

$$\Gamma_H = \frac{sin(\theta) - \sqrt{\epsilon - cos^2(\theta)}}{sin(\theta) + \sqrt{\epsilon - cos^2(\theta)}}, \tag{7}$$

$$\Gamma_V = \frac{\epsilon sin(\theta) - \sqrt{\epsilon - cos^2(\theta)}}{\epsilon sin(\theta) + \sqrt{\epsilon - cos^2(\theta)}}, \tag{8}$$

where $\epsilon = \epsilon_r - j60\lambda\sigma$ is the complex dielectric constant. The reflection coefficients are functions of the relative permittivity ($\epsilon_r$) and conductivity ($\sigma$) of the reflecting surfaces which account for the attenuation of the signal in the reflection process. They also depend on the angle of incidence ($\theta$). The reflecting surface in this study is assumed to be concrete, for which $\epsilon_r = 3$ and $\sigma = 2 \times 10^{-5}$ S/m in the GNSS frequency spectrum. The rotation matrices in Equation 6 rotate the coordinates to account for the change in polarization during reflection. Angle $\psi$ is the angle between the normal of the incident plane and the direction of the propagating signal from satellite to plane ($tx$) and from plane to receiver ($rx$) [26,50],

$$\psi_{tx} = tan^{-1}\left( \frac{||\hat{n} \times \hat{txp}||}{\hat{n} \cdot \hat{txp}} \right), \tag{9}$$

where $\hat{n}$ is the normal vector of the incident plane and $\hat{txp}$ is the unit vector between the transmitter and the reflecting point. Analogously, angle $\psi_{rx}$ can be defined.

The power of the compound signal which is the combination of the LOS and reflection MPC reads

$$P_{Compound} = P_{LOS}\sqrt{1 + 2\left( \frac{P_{MPC}}{P_{LOS}} \right)cos(\Delta\Phi) + \left( \frac{P_{MPC}}{P_{LOS}} \right)^2}, \tag{10}$$

where $P_{MPC}/P_{LOS}$ is the relative amplitude and $\Delta\Phi = tan^{-1}(Im(S_{MPC})/Re(S_{MPC}))$ is the relative phase of the multipath component.

### 4.5. The Role of Antenna Gain Patterns

The gain pattern of an antenna is a function of the antenna radiation efficiency and directivity. It shows how efficiently a transmitting antenna can convert electrical power to radio waves and, analogously, how a receiving antenna can convert radio waves into electrical power in different angles of departure and arrival. As it can be inferred from Equation (1), the power received at the receiving antenna is a function of the gain pattern of both transmitting and receiving antennas.

- Satellite

Table 2 shows the RHCP values of $G_t$ used in this study for GPS satellites. The gain pattern of the GPS satellites depend on the type of the space vehicle transmitting the PRN code [51]. The angle of departure (AoD) can be calculated from the *Off-Nadir* angle $\beta$:

$$\beta = sin^{-1}\left( \frac{R_{Earth}sin(El + 90°)}{R_{SV}} \right), \tag{11}$$

where $R_{Earth}$ is the radius of the Earth and $R_{SV}$ is the radius of the space vehicle from the center of the Earth. *El* denotes the elevation angle of the satellite. The Off-Nadir angle can change from $0°$ to $15°$, for which the corresponding value can be selected from Table 2. The LHCP values which are needed in Equation (2), can be calculated from

$$G_{lhcp} = G_{rhcp} \frac{-Ax + 1}{-Ax - 1}. \tag{12}$$

The axial ratio ($Ax$) of 1.2 dB is applied for GPS satellites. It can be denoted here that for GLONASS satellites, there are no exact gain patterns available to be used for this calculation; rather, some mean values, e.g., 14 dB for L1, can be used [51].

**Table 2.** RHCP Gain patterns used for the GPS satellite antenna $G_t$. Values are in dB.

| Block | | | | | | | | | | | | | | | |
|-------|------|------|--------|-------|-------|-------|------|-------|-------|-------|--------|-------|--------|-------|------|
| **IIF/III** | 13.75 | 13.8 | 14 | 14.3 | 14.51 | 14.75 | 15 | 15.45 | 15.6 | 15.7 | 15.8 | 15.7 | 15.55 | 15.25 | 14.8 |
| **IIR** | 13.0 | 13.2 | 13.45 | 13.6 | 13.9 | 14.25 | 14.5 | 15.0 | 15.1 | 15.3 | 15.3 | 15.15 | 14.85 | 14.4 | 13.5 |
| **IIRM** | 11.9 | 11.9 | 11.925 | 11.95 | 12.1 | 12.5 | 13 | 13.7 | 14.45 | 14.75 | 15.125 | 15.25 | 15.125 | 14.8 | 14.5 |

- Receiver

    The power received at the receiving antenna is attenuated based on the angle in which the signal impinges the antenna (the receiving antenna is assumed to be levelled). This attenuation depends on the gain pattern of the antenna, which usually degrades by decreasing the elevation angle. Here, the gain pattern is assumed to be only a function of elevation and not azimuth; thus, side lobes are neglected. Unfortunately, the gain pattern of the receiving antenna (Septentrio Altus NR3) is not known, and the gain pattern of a GNSS 800 series antenna is used. For evaluating the performance of the Altus receiver, 24 h data are collected in an open sky situation. On the other hand, a 24 h simulation was also conducted to see the results for different elevation angles. Comparing the measured $C/N_0$ data with the theoretical $C/N_0$ calculated from a simulation offers the possibility to adapt the used gain pattern to better fit the real data. Therefore, a modified version of the gain pattern is introduced, and the simulation is conducted with this modified version as shown in Figure 9b.

*4.6. Carrier-to-Noise Density Ratio*

The carrier-to-noise power density ratio is the ratio of the power of the carrier signal ($C$) to the noise power ($N_0$) in a 1 Hz bandwidth expressed in dB-Hz.

$$C/N_0 \text{ [dB-Hz]} = 10log_{10}(P_S) - 10log_{10}(N_0), \tag{13}$$

where

$$N_0 = k \cdot T_E. \tag{14}$$

$k = 1.38 \times 10^{-23}$ is the Boltzmann constant and $T_E$ is the environment temperature in Kelvin. The average temperature for the date and time of the experiment is $20\,°C$ or 293.15 K. Therefore, the noise power density becomes $-203.9303$ dB-Hz.

Furthermore, if there is any attenuation due to diffraction or passing through foliage, the corresponding value will be reduced from Equation (12). For multipath situation, the compound power is considered, but for NLOS situations, only the multipath component which is the reflected part is taken.

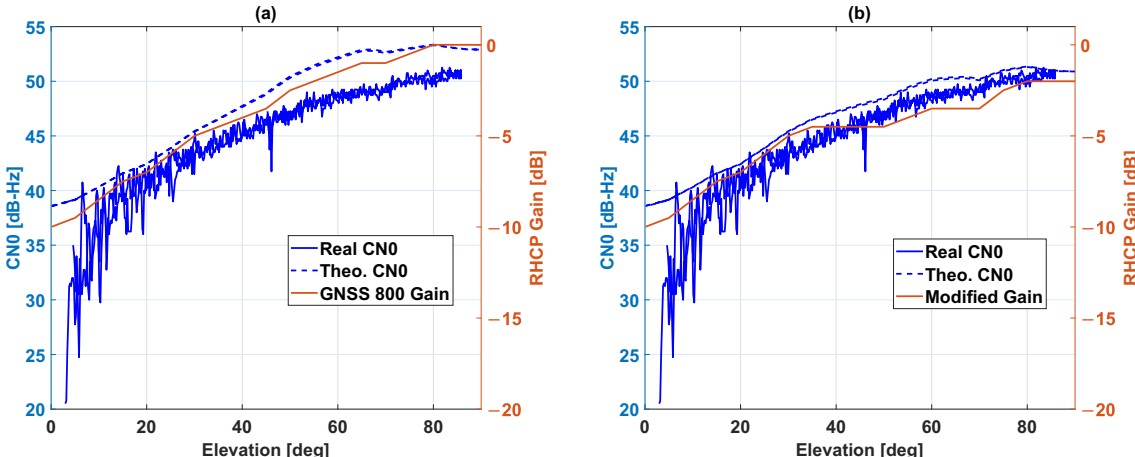

**Figure 9.** Investigation of the receiver antenna gain pattern (exemplary for high-elevated GPS PRN 1): (**a**) Real $C/N_0$ from the Altus receiver, theoretical $C/N_0$ from simulation using gain pattern of GNSS 800 series antenna, and the gain pattern of the GNSS 800 series antenna. (**b**) Real $C/N_0$ from the Altus receiver, theoretical $C/N_0$ from simulation using modified gain pattern and the modified version of the gain pattern of the GNSS 800 series antenna.

### 4.7. Diffraction

Diffraction loss ($L_D$) is calculated using the Knife Edge Diffraction (KED) model [52],

$$L_D(v) = \frac{1}{1-j}\left(F(v) + \frac{1-j}{2}\right), v = b\sqrt{\frac{2}{\lambda r_2}},\tag{15}$$

in which $F(v)$ is the Fresnel integral, b is the deviation of the edge of the building from LOS, and $r_2$ is the distance from the diffracting edge to the receiving antenna [53]. The diffraction loss is then reduced from the nominal value to obtain the $C/N_0$ value. It can be mentioned here that in this study, the diffraction is calculated considering the third Fresnel zone [41].

### 4.8. Foliage

It is probable that the signal radiating from the satellite passes through foliage; hence, they are further attenuated than the unshadowed signals [54–56]. Specifically in our case, while evaluating the $C/N_0$ along the trajectory, we noticed that before way-point C (Figure 2), especially in the first six rounds, the solution proceeds to float or code mode. Looking at the visibility status of PRNs 3 and 21, which have medium elevation, we noticed that they were in an LOS situation, meaning that there was no obstruction by any building. Moreover, we can be sure that there was no blocking building from the high-elevated satellites, which also have a reduction in signal strength in this point. Examining the position, we saw there were three consecutive trees whose canopies covered the street, mainly on the southern side of the street (Figure 10). There are other trees affecting the signal reception in all parts of the trajectory except between way-points D to F. These trees are from very different types and with different geometrical shapes.

The 3D model used for ray tracing does not include any 3D model for trees. Therefore, we added a rough rectangular cuboid to the 3D model (Figure 10). The model for foliage attenuation explained in [57] assumes the tree as a slab in which the leaves are thin disks, and branches are thin cylinders randomly distributed inside the slab. In Figure 4 of this study, the attenuation for different frequencies is plotted. At a 90° incident angle, the attenuation for 900 MHz is 7.1 dB/m. The corresponding value for 1.5 GHz is 1.15 dB/m. In Figure 5 of this study, the attenuation is plotted against the incident angle of the signal and the slab for the frequency of 900 MHz.

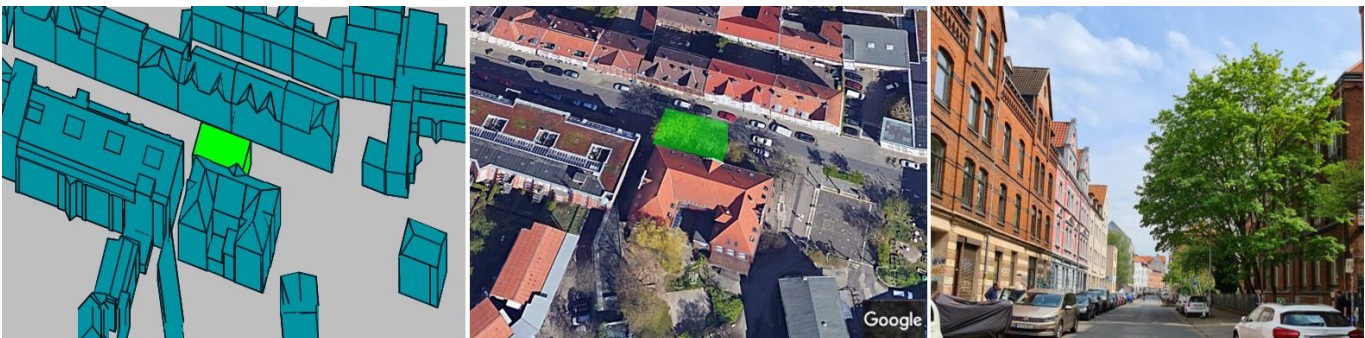

**Figure 10.** Real position of one of the trees (**right** and **middle**) and the 3D model used to predict the signal fading by the foliage (**left**).

A polynomial can be extracted for attenuation (dB/m) for 1.5 GHz, and the power loss can be calculated by

$$L_F = (0.000136\theta^2 - 0.02468\theta + 2.2696)d_F,\tag{16}$$

with $L_F$ attenuation caused by foliage in dB, $\theta$ being the incident angle in degrees and $d_F$ being the path length through foliage slab in meters.

The flowchart in Figure 11 summarizes the required inputs and the sequence of calculating needed parameters in order to predict the $C/N_0$, as it was explained in detail throughout this section.

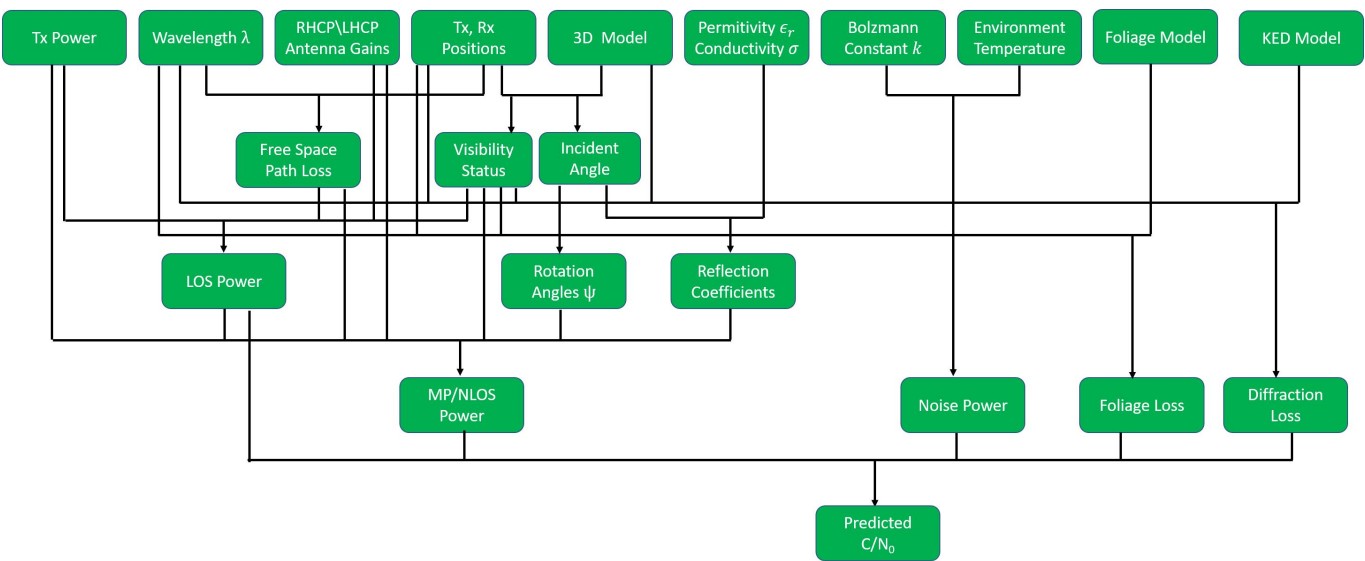

**Figure 11.** Flowchart summarizing the steps to predict $C/N_0$.

## 5. Prediction Results

### 5.1. $C/N_0$ Prediction

In Figures 12 and 13, the predicted $C/N_0$, as well as the measured ones, are depicted for six different GPS PRNs. In each of the $C/N_0$ time series, the color code at the bottom shows the solution status (cf. Figure 2), and the color code around 10 indicates the status of the signal as expressed in Figure 6. Furthermore, tree intersection, the possibility that the signal be reflected more than once, and the tracking status of the phase signal in the receiver are also included in this bar.

Figure 12a shows the results for PRN 3 which has an azimuth of $-100°$ and an elevation of $74°$ (cf. Figure 12d). For this PRN, the trees are correctly detected before C and after J, and the corresponding attenuation is calculated. As it is quite high-elevated, it is generally in an LOS condition with some diffraction parts which are correctly aligned with

measured data. Figure 12b shows the results for PRN 4 with an azimuth of $-170°$ and an elevation of $23°$ (cf. Figure 12d). It is obstructed in many parts of the trajectory. In some blocked areas, diffraction is predicted which seems realistic ,e.g., between points I and J. While in some cases, e.g., between G and H, it is predicted as blocked, and also no phase is tracked, there are some low $C/N_0$ values. After point J, a dense block of trees, obscures the signal completely. In addition, the spike at point K can be attributed to a tree. Figure 12c depicts the predicted and measured $C/N_0$ for PRN 17 which has an azimuth of $-68°$ and an elevation of $44°$ (cf. Figure 12d). The overall level of predicted values is in the range of measured values, especially in LOS situations. The simulation correctly shows no decrease in $C/N_0$ for foliage before point C, while other spikes because of trees, are well detected and evaluated. The decrease after point D up to before F is simulated by the NLOS status, but the level of the signal is significantly decreased up to 15 dB-Hz, whereas simulation shows a decrease of maximum 5 dB-Hz. At the point of drop, being tagged by diffraction, a decrease of 13 dB-Hz can be seen. Exactly at the point of $C/N_0$ drop, the phase signal is interrupted.

Figure 13a depicts the results for PRN 19 with an azimuth of $-45°$ and an elevation of $26°$ (cf. Figure 13d). This PRN is tracked in a blocked area in the diffraction mode between B and C. Here, also a sudden decrease in $C/N_0$ stops phase tracking. Between F and G, some consecutive trees make teeth shape, and another tree is detected just before G, where the attenuation is predicted much lower than the real data. In continue, some spikes after H, after I, at J and K, are well detected as trees, but the values are not completely aligned with real data or slightly displaced. This could be due to the limited accuracy of the tree models. Figure 13b shows predicted and measured $C/N_0$ for PRN 22 which has an azimuth of $75°$ and an elevation of $81°$ (cf. Figure 12d). This PRN has a high elevation, and the phase signal istracked continuously. Figure 13c illustrates $C/N_0$ results for PRN 32 with azimuth and elevation of $44°$ and $18°$, respectively (cf. Figure 12d). PRN 32 has a low elevation and is rarely in an LOS situation. After points D to F and also between G and H, the code is tracked that could be from multi-reflection. The phase signal is only partly tracked, mainly in LOS situations. Some spikes after K, are well detected because of foliage.

### 5.2. Predicted Measured Difference

Figure 14 illustrates the histogram of the differences between predicted and measured $C/N_0$ for the four different visibility status. The differences  for LOS and MP are, in most cases, far below 10 dB-Hz, and a mean offset of 1.78 dB-Hz and 3.21 dB-Hz is found, respectively, cf. Table 3. Taking into account that (i) the actual antenna gain pattern is only approximately known and the side lobes are not considered and that (ii) some trees and details may still be missing in the model, the general $C/N_0$ prediction fits well the real data of the one commercial receiver it was compared with. Considering the typical $C/N_0$ scatter of approximately 1 db-Hz at higher elevations, the obtained mean offsets are within this order of magnitude.

For NLOS situations, the simulation is less performant; this is also true for "blocked" cases. Here, the behavior of the other receivers has to be studied. In addition, the $C/N_0$ for the multi-reflection case which is partially contained in both cases must be taken into account.

**Table 3.** The mean value of the difference between predicted and real $C/N_0$ in Round 5 for four different visibility status. The modified gain pattern is used for these calculations.

| Visibility Type | LOS | MP | NLOS | BLKD |
|---|---|---|---|---|
| Mean of Difference [dB-Hz] | 1.78 | 3.21 | 11.90 | 8.74 |

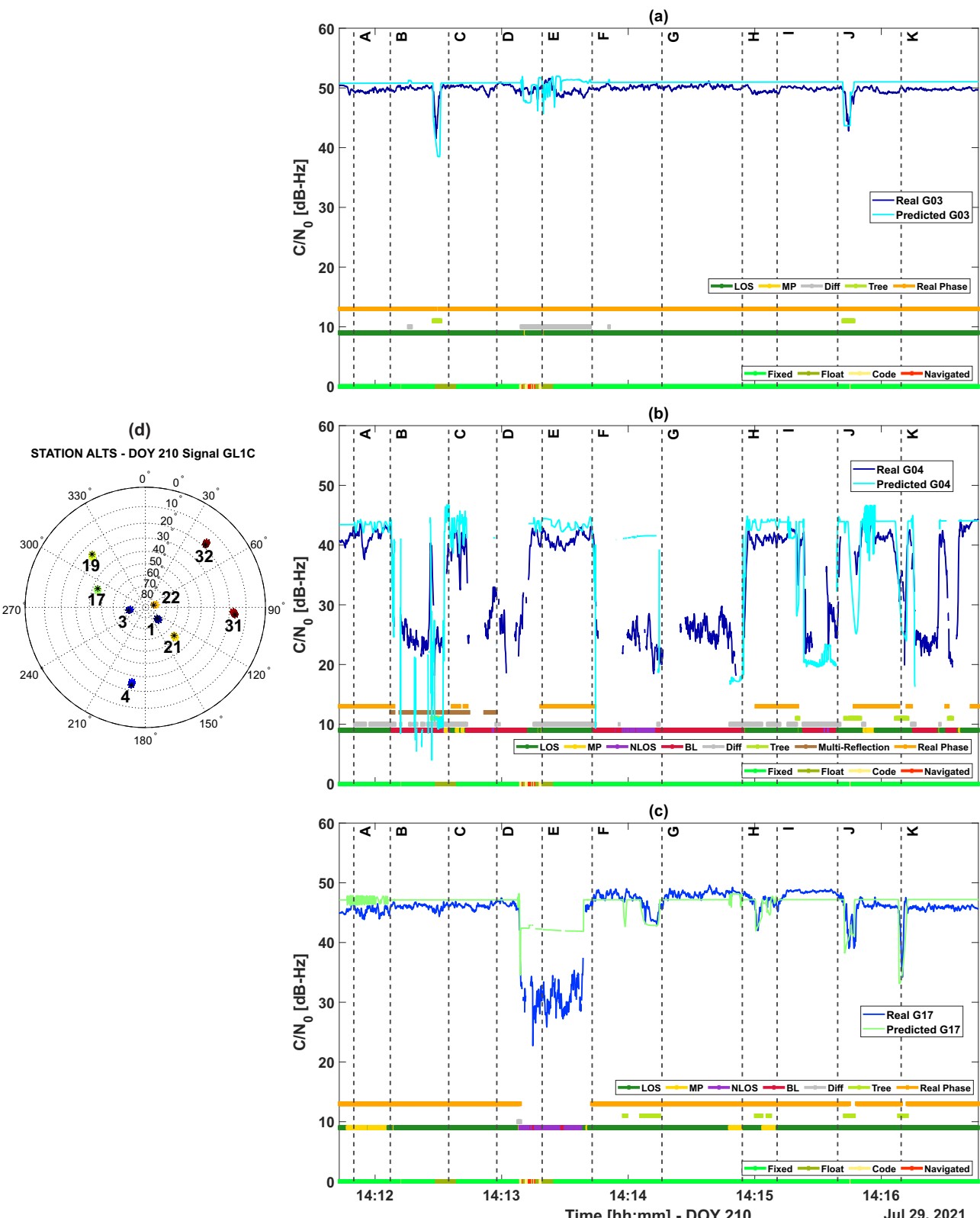

**Figure 12.** The observed and predicted $C/N_0$ in Round 5 for GPS constellation: (**a**) PRN 3, (**b**) PRN 4, (**c**) PRN 17. The lower bar (at 0) shows the solution status (cf. Figure 2a) and the upper bar (at 10) indicates the visibility status (cf. Figure 5). The way-points A to K in subfigures a, b and c, are as defined in Figure 1a. (**d**) Shows the sky plot of the GPS satellites during the round.

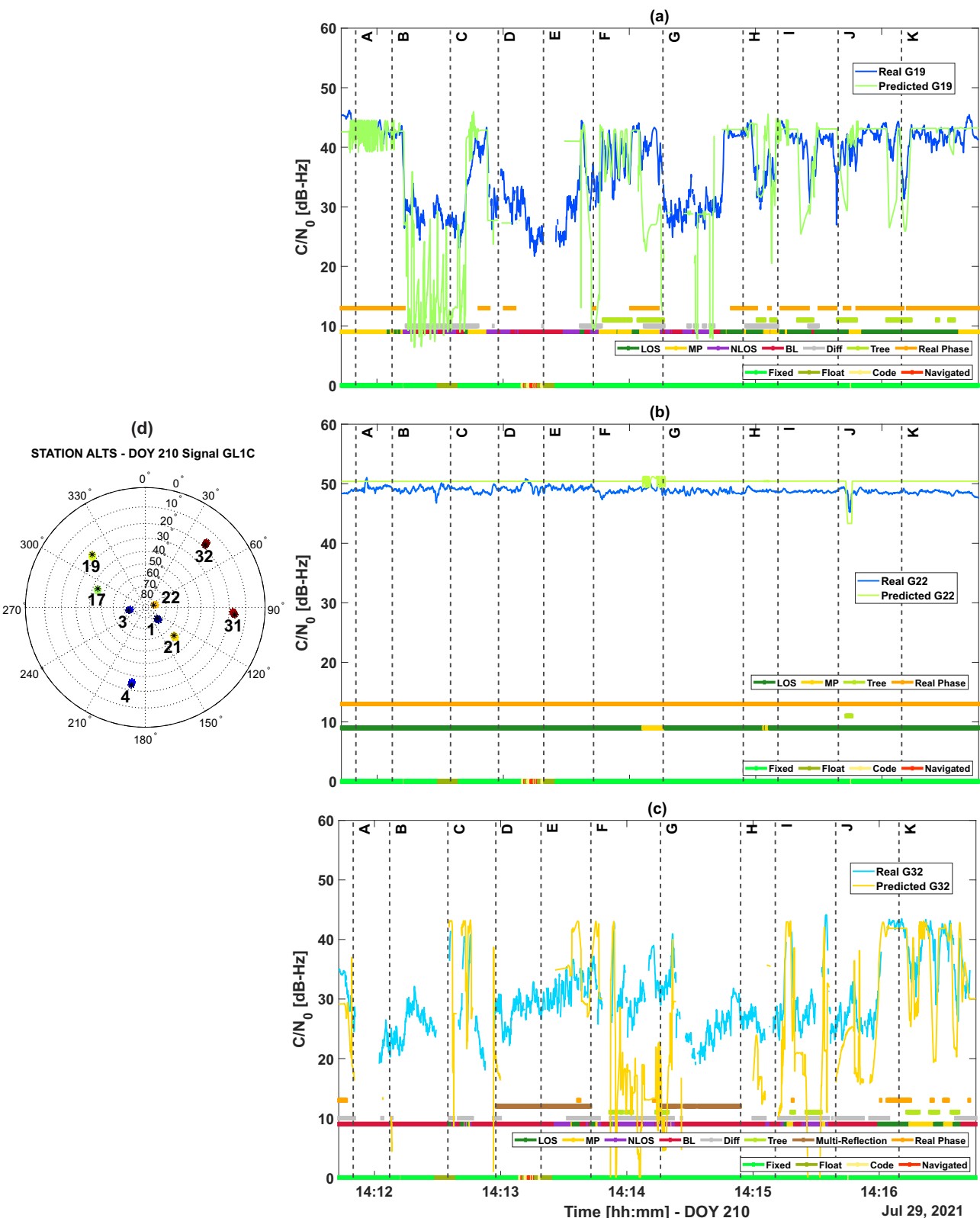

**Figure 13.** The observed and predicted C/N$_0$ in Round 5 for GPS constellation. (**a**) PRN 19, (**b**) PRN 22, (**c**) PRN 32. The lower bar (at 0) shows the solution status (cf. Figure 2a) and the upper bar (at 10) indicates the visibility status (cf. Figure 5). The way-points A to K in subfigures a, b and c, are as defined in Figure 1a. (**d**) Shows the sky plot of the GPS satellites during the round.

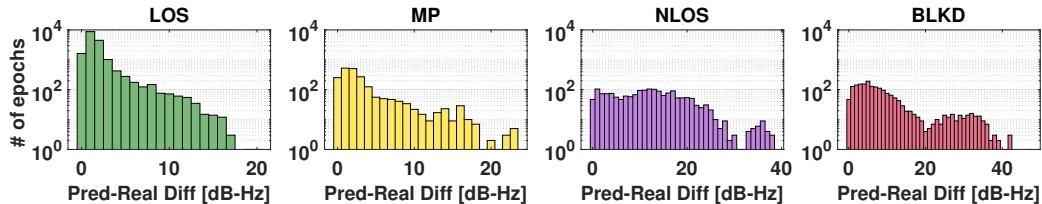

**Figure 14.** Histogram of the difference between the predicted and real $C/N_0$ values for four different visibility statuses.

### 5.3. Adjusting the Visibility Based on Prediction

In Section 4.2, we see that the tracking of the phase observation is different from that of code observations. Looking at the phase observations from measured values by the receiver, we can detect a $C/N_0$ threshold of around 30 dB-Hz for the tracked phase. Now, we have some predictions for $C/N_0$, so we are able to apply such a threshold to our predictions to modify the visibility prediction (cf. Section 4.2). Table 4 shows the confusion matrix for the GPS phase observations applying a threshold of 30 dB-Hz.

**Table 4.** Adjusted confusion matrix with the $C/N_0$ threshold of 30 dB-Hz. The green color indicates true positive and true negative, the red color indicates false positive and false negative. Numerical values are in percent only for predicted satellites.

|   |   | **Phase** | |
|---|---|---|---|
|   | % | Real Visible | Real Not Visible |
| **GPS** | Predicted Visible | 59.14 | 10.47 |
|   | Predicted Not Visible | 0.66 | 29.73 |

Comparing the adjusted values to those of Table 1, it can be observed that the true negative noticeably increased and the false positive decreased.

## 6. Conclusions

For predicting the integrity of NRTK positioning in the framework of an optimum path planning, we need to better predict the quality of the observations. In this contribution, we investigate the prediction of the most important quality measure, i.e., the carrier-to-noise density ratio $C/N_0$. For this purpose, we use a ray-tracing algorithm combined with a 3D building model, a simple channel model to predict the signal availability, as well as attenuation due to the antenna gain pattern, diffraction effects, signal reflections or propagation through foliage. Considering the partial knowledge of the antenna gain pattern as well as shortcomings of the building model like incompleteness in needed details as well as computational intensive operations especially for investigations beyond single reflection events, the general $C/N_0$ prediction fits well the real data of the one commercial receiver compared with. Taking the typical $C/N_0$ scatter of approximately 1 dB-Hz at higher elevations into account, the obtained mean offsets of 1–3 dB-Hz for classes LOS and Multipath are in a good agreement. The time series of the predicted $C/N_0$ indicate most of the abrupt $C/N_0$ changes correctly, which coincides with a loss of the carrier phase signal and thus the absence of a reliable RTK solution.

Further investigations will focus on the impact of predicted signal deterioration on the positioning results. To this end, a linearized Kalman Filter will be used to predict integrity parameters, i.e., position error and protection level.

**Author Contributions:** Conceptualization and methodology, A.K. and S.S.; software, A.K.; writing—original draft preparation, A.K.; writing—review and editing, S.S. All authors have read and agreed to the published version of the manuscript.

**Funding:** This research was funded by the German Academic Exchange Service (DAAD) Graduate School Scholarship Program (GSSP) number 91750240 as part of the DFG funded research training group i.c.sens. The publication of this article was funded by the Open Access Publishing Fund of Leibniz Universität Hannover.

**Data Availability Statement:** The data presented in this study are available on request from the corresponding author.

**Acknowledgments:** The authors would like to thank Anat Schaper for providing the diffraction information. Tobias Kersten supported and assisted for the data acquisition.

**Conflicts of Interest:** The authors declare no conflict of interest.

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
