# Peer review of "Predicting C/N0 as a Key Parameter for Network RTK Integrity Prediction in Urban Environments"

_remotesensing, doi:10.3390/rs15194850_

Round 1
Reviewer 1 Report
GNSS signals are very important for intelligent vehicle localization system, this paper investigate the predictability of the GNSS signal strength, which is meaningful for the usage of GNSS signals. The paper is well organized and written. However, the following aspects should be further considered to improve the manuscript:
1. In the abstract, it is said that "As GNSS real time kineamtic (RTK) solutions are needed to ensure lane lavel accuracy", actually, in the applications in the autonomous vehicles, we usually require the accuracy in RTK mode to be centimeter-level, so that the final integrated localization system could reach lane level accuracy while losing the RTK signals. By the way, "lavel" is a typo
2. The RTK is commonly used in the localization systems in the autonomous vehicles, so some applications should also be introduced in the introduction section. For example, you could include the following papers to improve the literature review: improved vehicle localization using on-board sensors and vehicle lateral velocity, an automated driving systems data acquisition and analytics
platform, automated vehicle sideslip angle estimation considering signal measurement characteristic
3.The limitation of the current work and future work should also be discussed.
Author Response
- Thanks a lot for the review. we updated the corresponding part in the abstract so that even a higher accuracy is needed.
- Thanks for the suggestions. Although they are related to autonomous vehicles localization, we found the topics a bit far from the main focus of the paper.
-
The current method uses the 3D building model, which has its own shortages. In some cases, the 3D models are not complete in details. In addition, working with 3D models consumes lots of time, so, it has to be optimized in this direction.
In continue, we try to investigate the impact of our predictions up to now, on the positioning results by developing a linearized Kalman Filter. To see how the integrity parameters i.e. position error and protection level can be predicted.
We clarified these points in the conclusion section.
Reviewer 2 Report
Very interesting and meaningful result and approach. Quality of presentation is high, especially the experiment details.
From line 225 to 232, the article give the needed information for satellite and receiver coordinates, however, it should be time dependent. Here mentioned tight couple GNSS with IMU, it is from experiment, and is post-processing? This article is focus on integrity prediction based on route planning? So the trajectory should be pure planning, and time parameter is varying? please clarify. Because I just wonder if it is about real-time prediction, we may use the prediction results exclude satellite with worse C/N0, and improve the positioning?
In line 49, the abbreviation 3DMA may need several words to explain?
Author Response
1. Thank you for the review. Yes, in this experiment the reference trajectory is calculated in post-process using tightly coupled GNSS+IMU data. This reference trajectory is used here as the predicted trajectory to facilitate the comparison. Ruwisch and Schön (2022) ITM, (2023) ITSC showed that the classifications are typically preserved in a road section of a few meters.
In real application, the prediction should be calculated e.g. from map services. The prediction of satellite visibility and further predicting the C/N0 can be made based on this predicted (approximate) trajectory, the known satellite almanac as well as the known 3D building model. As mentioned in section 5.3, the visibility status can be improved by excluding the satellites with worse C/N0 bellow a phase tracking threshold (as you mention).
We add some clarifications accordingly in the text.
2. Thanks, we made changes accordingly. It has been explained some lines later.